# COVID-19-Specific Mortality among World Trade Center Health Registry Enrollees Who Resided in New York City

**DOI:** 10.3390/ijerph192114348

**Published:** 2022-11-02

**Authors:** Janette Yung, Jiehui Li, Rebecca D. Kehm, James E. Cone, Hilary Parton, Mary Huynh, Mark R. Farfel

**Affiliations:** 1New York City Department of Health and Mental Hygiene, World Trade Center Health Registry, New York, NY 11101, USA; 2Department of Epidemiology, Mailman School of Public Health, Columbia University, New York, NY 10032, USA; 3New York City Department of Health and Mental Hygiene, Bureau of Communicable Diseases, New York, NY 11101, USA; 4New York City Department of Health and Mental Hygiene, Bureau of Vital Statistics, New York, NY 10013, USA

**Keywords:** COVID-19, mortality, World Trade Center disaster, racial disparity, health inequity

## Abstract

We examined the all-cause and COVID-19-specific mortality among World Trade Center Health Registry (WTCHR) enrollees. We also examined the socioeconomic factors associated with COVID-19-specific death. Mortality data from the NYC Bureau of Vital Statistics between 2015–2020 were linked to the WTCHR. COVID-19-specific death was defined as having positive COVID-19 tests that match to a death certificate or COVID-19 mentioned on the death certificate via text searching. We conducted step change and pulse regression to assess excess deaths. Limiting to those who died in 2019 (*n* = 210) and 2020 (*n* = 286), we examined factors associated with COVID-19-specific deaths using multinomial logistic regression. Death rate among WTCHR enrollees increased during the pandemic (RR: 1.70, 95% CL: 1.25–2.32), driven by the pulse in March–April 2020 (RR: 3.38, 95% CL: 2.62–4.30). No significantly increased death rate was observed during May–December 2020. Being non-Hispanic Black and having at least one co-morbidity had a higher likelihood of COVID-19-associated mortality than being non-Hispanic White and not having any co-morbidity (AOR: 2.43, 95% CL: 1.23–4.77; AOR: 2.86, 95% CL: 1.19–6.88, respectively). The racial disparity in COVID-19-specific deaths attenuated after including neighborhood proportion of essential workers in the model (AOR:1.98, 95% CL: 0.98–4.01). Racial disparities continue to impact mortality by differential occupational exposure and structural inequality in neighborhood representation. The WTC-exposed population are no exception. Continued efforts to reduce transmission risk in communities of color is crucial for addressing health inequities.

## 1. Introduction

The severe acute respiratory syndrome coronavirus-2 (SARS-CoV-2 or COVID-19) pandemic emerged as a global health emergency in the beginning of 2020, resulting in large-scale physical and mental health concerns as the world was confronted with the unprecedented crisis. SARS-CoV-2, the virus that causes COVID-19, may cause severe respiratory complications that lead to life-threatening situations and require invasive medical interventions. Excess mortality from direct and indirect causes of the pandemic has been well reported in different countries [1,2,3]. New York City (NYC) was an early epicenter of the pandemic, with more than 6000 residents testing positive and over 500 deaths per day during the peak period in March and April of 2020 [4]. The COVID-19 pandemic has disproportionally affected communities of color, the elderly, and those with pre-existing chronic health conditions, such as diabetes, obesity, chronic obstructive pulmonary disease, cardiovascular diseases, hypertension, and chronic kidney disease, resulting in more deaths among these individuals [5,6,7,8].

In addition to the direct impact of COVID-19 on physical health, mental health conditions such as depression, anxiety, and post-traumatic stress disorder (PTSD) were also reported to increase [9,10] during the pandemic. Many individuals also experienced indirect economic and social consequences such as unemployment, loss of income, closures of schools and universities, increase in domestic violence, child neglect or abuse, and other financial insecurities that contribute to poor mental health [9,11,12]. There were also suggestions that the suicide rate rose as an indirect consequence of COVID-19’s socioeconomic impact, which is becoming a more pressing concern as the pandemic continues [13]. Another major indirect cause of death stemming from the COVID-19 pandemic was the disruption of health care. Those with either pre-existing or newly diagnosed physical or mental health conditions may have experienced delays in receiving a medical diagnosis and treatment secondary to the reduction in public services or avoiding care due to concerns about contracting COVID-19 [14]. Hence, lack of hospital beds and other treatment delays may have increased the risk of mortality in those with already compromised health [7].

Many individuals present in lower Manhattan during the 11 September 2001 World Trade Center (WTC) attacks witnessed or experienced traumatic events, consequently resulting in significant physical and psychological morbidities from the exposure [15], namely asthma, sinusitis, gastroesophageal reflux disease (GERD), PTSD, and depression [15,16,17]. Having multiple co-morbidities resulting from a higher level of exposure [17] is also an indication of poorer health-related quality of life [16,18]. Though no excess overall mortality in the WTC-exposed population was reported from 2003–2014, a dose–response relation of WTC-related exposure with increased risk of all-cause mortality has been reported [19]. Excess death due to certain types of malignant neoplasms, such as neoplasms of the lymphatic and hematopoietic tissues, were found to be higher among WTC survivors compared with the NYC population [19]. The suicide rate was also elevated in rescue/recovery workers compared with the general NYC population [19]. Moreover, those who suffered with probable PTSD resulting from the WTC exposure had a higher all-cause, cardiovascular, and external-cause mortality than those without PTSD [20]. Therefore, it is possible that the COVID-19 pandemic might have complicated or heightened the severity of the existing health conditions in this traumatized population, resulting in a higher risk of death than expected.

We aimed to examine whether there was excess all-cause and COVID-19-specific mortality among the World Trade Center Health Registry (WTCHR) enrollees who resided in NYC in 2020. We also examined the association of socioeconomic and behavioral characteristics with COVID-19-specific mortality compared with pre-pandemic mortality and non-COVID-19-specific mortality in 2020. 

## 2. Materials and Methods

### 2.1. Data Sources and Study Population

The WTCHR is by far the largest post-disaster registry in U.S. history. It was established in 2002 to conduct longitudinal follow-up on individuals who were exposed to the WTC disaster. The details of the Registry are described elsewhere [18,21,22]. In brief, the Registry completed enrolment in 2003–2004 and had recruited over 71,000 participants. The cohort includes rescue/recovery workers, lower Manhattan residents, office workers, students, and passerby. All enrollees provided verbal informed consent at enrolment (Wave 1), followed by completion of a detailed questionnaire related to sociodemographic characteristics, WTC exposure, and physical and mental health history. Since enrolment, four subsequent waves of surveys were completed in 2006–2007 (Wave 2), 2011–2012 (Wave 3), 2015–2016 (Wave 4), and 2020–2021 (Wave 5). In the present study, all data on the risk factors examined were from Wave 1 to Wave 4.

The WTCHR data were periodically linked to the NYC BVS for mortality surveillance through 2020. The NYC Department of Health and Mental Hygiene (DOHMH). Bureau of Vital Statistics (BVS) maintains the single largest repository of mortality data in NYC. All deaths that occur in NYC are required to be reported to the DOHMH within 24 h from the medical examiner who completes the person, time, location, cause of death and other necessary medical information of the decedent using an electronic reporting system, eVital, and within 72 h from the Morgue and funeral director for the rest of the death-related information. NYC BVS validated and registered the death reports upon receipt.

In the present study, the study population was from the WTCHR cohort and limited to those who resided in NYC based on the last known residential address. We excluded those who were presumably not living in NYC during the study period. Considering the discordancy between residency and receipt of health care or other social services in NYC, we used a consistent approach across all years to determine the residency status of our enrollees for a given year based on the last known address. The number of eligible enrollees per year ranged between 36,384 and 42,293 during 2015–2019 and was 37,651 in 2020.

### 2.2. Ascertainment of Death

Deaths occurring among eligible enrollees were identified via linkage to the BVS death certificates through probabilistic matching. Potential matches were records that matched to parts of key identifiers such as name, date of birth, social security number and last known address. Key identifiers from the Registry and data from the vital records were electronically compared, and a score was assigned for each potential match based on degree of similarity. These scores aided identification of deaths of enrollees during manual review by two independent reviewers. In the case of disagreement, a third reviewer determined whether the case was a match [23]. For the study period, linkage data for years 2019–2020 was received in early 2021 and includes both the confirmed and probable COVID-19-specific deaths occurred in 2020. Confirmed COVID-19-specific death was defined as having a positive molecular test for the virus that causes COVID-19 that matched to the death certificate and did not die of external causes such as gunshot wounds or drug overdose. To address instances in which a person was diagnosed with COVID-19 and survived, but later died, likely of other causes, starting in June 2020, people who died more than 60 days after their COVID-19 diagnosis and starting on 3 August 2021, people who died more than 30 days after their COVID-19 diagnosis who did not have “COVID” or similar listed on their death certificate were removed from the COVID19-specific death count. Since 14 April 2020, DOHMH also began to report probable COVID-19–specific deaths, when the death certificate mentioned COVID-19 or an equivalent term (e.g., COVID-19, SARS-CoV-2, or another term) as the cause of death in the absence of positive molecular test for the virus that causes COVID-19 [4]. Both confirmed and probable deaths are considered as COVID-19-specific death in this study.

### 2.3. Predictive Variables

Age at death, gender, race and ethnicity, last known marital status, income, smoking history, history of 9/11-related PTSD, neighborhood proportion of essential workers, type of enrollees, and having at least one chronic health condition were examined. Type of enrollees was obtained at enrollment and defined as either community members or rescue/recovery workers. All other variables, except for neighborhood proportion of essential workers, were defined based on the last known survey or location data of the decedents. Age at death was examined as a continuous variable in multivariable regression. Marital status was categorized as never married versus all other. Income was based on the last update status up until wave 4 and was categorized as ≤75,000 and >75,000. Smoking history was defined as either never or ever smoked, with those who reported to have smoked in the past in any of the wave surveys (Wave 1 to Wave 4) defined as ever smoked.

Differential occupation risk has been found to be associated with COVID-19 mortality, with essential workers enduring disproportionally higher mortality rates than non-essential workers [24]. The WTCHR does not have complete individual level data on occupation and so neighborhood proportion of essential workers by zip code from the 2019 American Community Survey’s 1-year estimate was evaluated instead [25]. Individual zip code was mapped with Zip Code Tabulation Area (ZCTA) from the U.S. Census. All the zip codes in our sample had the same value as ZCTA except for one, 10138, which was converted to 10016 for ZCTA based on the UDS crosswalk file [26]. We selected six major occupations of essential workers in the civilian population ≥16 years old that were highly associated with COVID-19 deaths [27]: (1) healthcare support, (2) protective service, (3) food preparation and serving related, (4) building and grounds cleaning and maintenance, (5) personal care and service, and (6) production, transportation, and material moving. The proportion of each occupation by zip code was categorized dichotomously using the median proportion of all NYC zips as the break point. A composite dichotomous variable of neighborhood proportion of essential workers was created with an event if the proportion of at least two of the six occupations by zip code was above median.

PTSD was assessed using the PTSD Checklist-Specific (PCL-S). Probable PTSD was defined as a score greater than 50. PCL-S is a 17-item self-reported measure of PTSD symptoms based on the American Psychiatric Association’s DSM-IV criteria and has demonstrated good validity [28,29]. It asked about symptoms in relation to a specific “stressful experience” in this case, the WTC attacks. A total symptom severity score was derived by summing the scores from the items. Based on the PTSD classification at all waves, we grouped individuals into two mutually exclusive categories: (1) never had PTSD and (2) history of PTSD.

Chronic health conditions included asthma, hypertension, angina, heart attack, coronary or other heart disease, stroke, and diabetes. Those who reported having any of these conditions in any of the primary wave surveys (wave 1 to wave 4) were considered to have at least one chronic health condition.

### 2.4. Statistical Analyses

The monthly number of deaths in 2020 and the mean monthly death count between 2015 and 2019 were enumerated in our sample. We compared the observed crude monthly total death rate per 1000 persons in year 2020, with the average monthly crude death rate per 1000 persons for the previous 5 years (2015–2019). To assess the excess mortality, we first used the difference-in-differences approach by comparing the observed monthly death count in year 2020 with the mean monthly death count in the previous 5 years, for with (all-cause) and without COVID-19-specific deaths for year 2020, then we examined the impact of the pandemic on the monthly death rate since March 2020 by conducting a single step change (sustained) and multiple individual pulse (temporary) quasi-Poisson time series regression models, with March to December 2020 as the COVID-19 pandemic intervention period, and individual bimonthly COVID-19 periods since March 2020 as the pulse periods, using RStudio version 4.2.1 (RStudio Team (2019). RStudio: Integrated Development for R. RStudio, Inc., Boston, MA, USA). Moreover, we limited our sample to those who died in 2019 and 2020 and divided the time into two periods: prior to pandemic (before March 2020) and since pandemic (March 2020 and onward), to assess the association of age, race/ethnicity, marital status, neighborhood proportion of essential workers, income, smoking status, history of PTSD, type of enrollees and any chronic health conditions with a three-level outcome: death prior to pandemic, non-COVID-19-specific death during the pandemic, and COVID-19-specific death by multinomial logistic regression model, using SAS version 9.4 (SAS Institute Inc., Cary, NC, USA).

## 3. Results

Overall, 1283 deaths were identified among an average of approximately 35,000 NYC enrollees between 2015 and 2020, amongst whom 210 and 286 individuals died in 2019 and 2020, respectively. Eighty-five confirmed and probable COVID-19-specific deaths were identified. A temporary increase in the death rate was observed in March and April 2020 compared with the average death rate for the same months in the previous 5 years (Figure 1).

Table 1a shows the results of the difference-in-difference analysis with (all-cause) and without COVID-19-specific deaths. There was no significant difference in the mean number of deaths for all-cause (β = 9.18, *p*-value = 0.17) and non-COVID-19-specific deaths (β = 0.68, *p*-value = 0.86) between pre- and post-pandemic periods by months (January and February 2020, March to December 2020, separately).

Table 1b shows the results of step change segmented regression models for the entire pandemic period (March 2020 onward), March and April 2020 only, and May to December 2020, specified as the intervention period in separate models. There was a significant overall increase in the rate of death in the pandemic period (RR: 1.70, 95% CL: 1.25–2.32), which is driven by the pulse in March and April (RR: 3.38, 95% CL: 2.62–4.30). From May to December 2020, there was no significant increase in the rate of death for any individual bi-monthly pulse (The combined result for May to December is presented in the last panel).

Table 2 compares the characteristics of WTCHR enrollees who died in year 2019 and 2020 by period (prior to vs. since the pandemic) and cause of death (all-cause and COVID-19 specific-death). The proportion of COVID19-specific deaths compared with non-COVID19-specific deaths differed by race/ethnicity, having a history of at least one chronic health condition, and ZCTA proportion of essential workers during 2019–2020 and March-December 2020 (*p*-values < 0.05). There was a higher proportion of all-cause deaths in 2020 vs. 2019 in enrollees living in a ZCTA above the median proportion of essential workers compared to at or below the median. 

In multinomial regression analyses (Table 3), being non-Hispanic Black and having a history of chronic health condition were significantly associated with COVID-19-specific deaths after adjusting for other individual level sociodemographic and behavioral factors (2.43, 95% CL: 1.23–4.77; AOR: 2.86, 95% CL: 1.19–6.88, respectively), but without adjusting for ZCTA proportion of essential workers. After further adjustment for ZCTA proportion of essential workers, only a history of chronic diseases remained statistically significantly associated with COVID-19-specific deaths (AOR:2.79, 95% CL:1.16–6.74).

## 4. Discussion

### 4.1. Main Findings

An increase in all-cause mortality during the peak of the pandemic in March and April 2020 was observed among WTCHR enrollees compared to pre-pandemic period. Although the mortality rate was lower than that in the overall NYC population, COVID-19-specific mortality drove about a three-fold increase in deaths in March–April 2020 compared with the previous 5-year average of our cohort, while a seven-fold increase was previously observed among the general population in NYC [30]. The risk factors associated with COVID-19-specific deaths are similar to those found in other populations, with having at least one chronic health condition significantly associated with higher COVID-19-specific death rates. Being non-Hispanic Black was also associated with COVID-19-specific deaths before, but not after adjustment for neighborhood proportion of essential workers. This suggests that the racial disparity in COVID-19-specific mortality was to some degree explained by the differential occupational exposure of essential workers, demonstrating the role of longstanding structural inequality in perpetuating negative health outcomes.

### 4.2. Excess Mortality

We observed an almost 3-fold excess in all-cause mortality during year 2020 compared to the average of previous 5 years among our enrollees. The excess mortality primarily occurred between March–May 2020, the early stage of the pandemic, consistent with what was observed in the general population in NYC. However, the proportion of COVID-specific deaths was not as high in the WTCHR as the NYC population [30], which might be due to medical services available to WTCHR enrollees regardless of their private health insurance status. Specifically, members of our cohort can receive medical evaluation, monitoring, and treatment from the World Trade Center Health Program (WTCHP) [31,32], including 9/11-related medical services at no out-of-pocket cost. Moreover, the Treatment Referral Program Unit of the WTCHR actively refers enrollees to the WTCHP and facilitates their access to monitoring and treatment [33]. The lower proportion of excess deaths in the WTCHR might also be due to population differences. For example, our cohort consists of predominantly non-Hispanic White enrollees with a relatively higher socioeconomic status at baseline than the NYC general population at that time [34].

### 4.3. Pre-Existing Chronic Health Conditions

Statistically significantly higher prevalence of certain physical and mental health conditions has been reported among the WTC-exposed population, including asthma, lower respiratory symptoms, GERD, PTSD, depression, and certain types of autoimmune disease and cancer [16,17,35,36]. It is well-known that those with chronic health conditions are more likely than those without these conditions to develop life-threatening consequences from COVID-19 infection that require invasive medical intervention or even result in death [5,6,7,8]. NYC areas with a higher prevalence of pre-existing health conditions also had significantly higher death rates than areas with a lower prevalence [37]. Our finding of a positive association between chronic health conditions and COVID-19-specific-mortality, after adjustment of covariates including type of enrollees, supports findings from the general population. In our sample, among those who died of COVID-19, about 31% had at least three conditions versus about 24% for those who died prior to the pandemic. Continued close monitoring in people with underlying conditions is necessary to prevent COVID-19-specific deaths in this higher risk group.

### 4.4. Racial Disparity

A burden of elevated COVID-19-specific mortality and positivity rate in non-Hispanic Black communities has been reported in NYC, particularly among the residents of zip codes with a high proportion of lower socioeconomic/groups [37,38]. This is likely attributed to a compounding effect of systemic racism such as a concentration of lower income, living in a less affluent neighborhood with a higher poverty rate, under-resourced hospitals, and a more crowded and congregated environment [37,39,40,41]. A NYC study reported an observed disparity in COVID-19 positivity rate and number of licensed and ICU unit hospital beds in the majority of non-Hispanic Black areas [39]. Although increased risk of hospitalization in Black individuals has been reported in some studies [39,40,41], likely driven by a higher prevalence of comorbidities [42], lower in-hospital COVID-19-specific mortality has been observed in Black compared with White individuals in some high-ranked NYC hospitals [40,43]. It is important to acknowledge that the overall outcomes for non-Hispanic Black individuals were not necessarily less favorable than those for the other groups once they were admitted to the hospital [40,42,43,44,45], indicating a racial gap in accessing health care.

Previous studies reported that differential occupational exposure is one main contributing factor to racial disparity in COVID-19 associated outcomes [46,47]. Non-Hispanic Black residents, in addition to living in already resource-deprived neighborhoods, are more likely to work in industries that are not amenable to remote working, have less ability to physically distance themselves from others in the workplace, and have more frequent exposure to infections while at work compared to other racial groups [46,47]. Our finding that the association between race and COVID-19 mortality was attenuated and no longer statistically significant after adjusting for neighborhood proportion of essential workers supports these previous literatures and may implicate that systemic and structural inequality determine the choices of residence. The high proportion of non-Hispanic Black living in areas with high proportion of essential workers may lead to the higher incidence of COVID-19 infections among them [37], either from direct exposure to the virus itself through work or indirect exposure from living in an environment where infectious agents are concentrated, rendering them and those they live with disproportionately vulnerable to SARS-CoV-2 exposure. Future studies accounting for this socioeconomic gap will further delineate the impact of racial disparity in COVID-19-associated health outcomes.

### 4.5. Strength and Limitations

Our study is not without limitations. First, we could not discount the potential misclassification of confirmed COVID-19-specific deaths in the early period of pandemic. In early March 2020, only hospitalized COVID-19 cases were getting tested due to limited testing availability, and reporting of probable COVID-19-specific deaths did not begin until 14 April 2020 [4]. It is possible that some early period COVID-19-confirmed deaths were missed as physicians may not have acquired enough knowledge of the complete clinical spectrum of COVID-19 and omitted it on the death certificate. However, our preliminary findings are consistent with NYC mortality trends related to COVID-19 and the issue of under-reporting likely results in underestimation of the association. Additionally, the early definition of COVID-19 mortality might have over-estimated the actual number of COVID-19-specific deaths by including incidental positive findings not associated with the actual cause of death. Despite this drawback, a significant increase in COVID-19-specific death that is comparable to the trend in NYC general population was observed among the WTCHR enrollees, deeming the impact of this bias subtle. Our relatively short post-pandemic follow-up period did not allow detection of any long-term secondary impact of the pandemic on mortality outcomes. Due to the limited number of deaths, we had to aggregate the data into a larger time unit (month vs. day), which resulted in fewer data time points. However, we used the five years prior to the pandemic as our baseline comparison group, and our approach of segmented time series is ideal for detecting any short-term impact of an explanatory variable of interest. Although this method accounts for factors temporally but not causally associated with death rates, we used multiple analytical approaches including visualization to look at the COVID-19-specific mortality among our sub-groups and the findings were similar. Another limitation is that we received the data from NYC BVS in early 2021, which did not include external deaths in the later period of 2020 that required investigation from the Office of Chief Medical Examiner. We also did not have the data on number of people infected among our enrollees, which limits our ability to evaluate the number of deaths relative to the number of people infected with COVID-19. We were only able to examine the overall mortality to understand the impact of COVID-19 pandemic on our enrollees. However, the reporting of all-cause excess deaths might provide a better measure of the impact of the pandemic, since the counting of confirmed and probable COVID-19 deaths might not include deaths among persons with SARS-CoV-2 infection who did not access diagnostic testing, tested falsely negative, or became infected after testing negative, died outside of a health care setting, or for whom COVID-19 was not suspected by a health care provider as a cause of death. It also does not include deaths that are not directly associated with SARS-CoV-2 infection [30]. In addition, our information on pre-existing health conditions was based on self-report and is not validated with clinical documentation. The true impact of existing chronic health conditions might have been underestimated as we only used the last known data. However, there has been a high level of agreement on co-morbidities with previous studies of the Registry, supporting the reliability of our findings. One strength of this study is that we collected data on demographics and health status from follow-up surveys since enrolment in 2003–2004, which enabled us to make causal inferences about COVID-19-specific deaths in the 9/11 exposed population due to the nature of the study design. Lastly, it was possible that other infectious agents were circulating during the overlapping 2019–2020 influenza season. We attempted to control for the long-term trend by including indicators of monthly interval or fitting Fourier terms in our time series regression equation to address for seasonality. However, we did not observe any differences from our main findings.

## 5. Conclusions

Among WTCHR enrollees, the COVID-19 pandemic led to a sharp rise in total deaths in March–April 2020, which returned to baseline afterwards. Though deaths across all NYC populations spiked more dramatically than among WTCHR enrollees at that time, our results were consistent with the general population. COVID-19-specific deaths were associated with being non-Hispanic Black and having at least one chronic health condition. Moreover, the neighborhood proportion of essential workers was found to be a major driver of the observed racial disparities in COVID-19-specific mortality. Structural racism continues to exert an impact on health and mortality outcomes despite ongoing efforts to reduce health inequity in vulnerable populations, and this disparity does not exclude those who were exposed to the 9/11 disaster. Given our nation’s long-standing history of racial disparity, continued public health policy to improve health outcomes in communities of color and protect essential workers is crucial to mitigate COVID-19-specific mortality and reduce health inequities.

## Figures and Tables

**Figure 1 ijerph-19-14348-f001:**
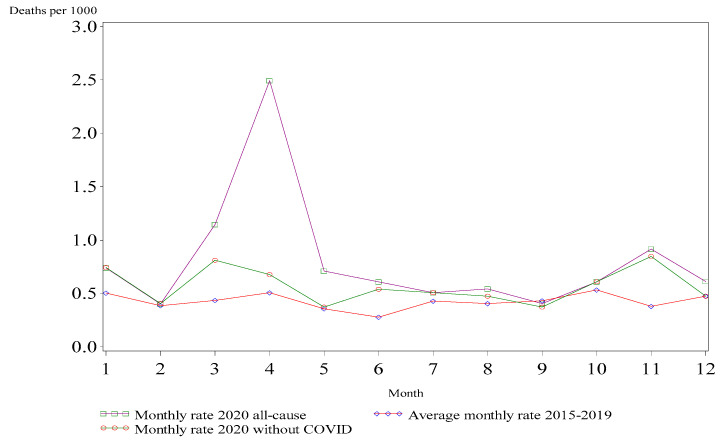
Average monthly crude death rate in 2015 to 2019 vs. monthly crude death rate in 2020 with (all-cause) and without COVID-19-specific deaths per 1000 persons among WTCHR enrollees.

**Table 1 ijerph-19-14348-t001:** (**a**). Difference-in-difference analysis in monthly number of all-cause (including COVID-19) and non-COVID-19-associated deaths. (**b**). Comparison of time series analysis with step change for the entire COVID-19 period and pulse periods in 2020.

(**a**)									
		**All-Cause**				**Non-COVID-19**			
		**Co-Efficient**	***p* Value**	**95% CI**		**Co-Efficient**	***p* Value**		**95% CI**
Constant		17.00	<0.01 *	12.08–21.92		17.00	<0.01 *		14.18–19.82
DID (year × month) ^1^		9.18	0.17	−4.03–22.39		0.68	0.86		−6.88–8.24
Year (2020 vs. 2015–2019)		0.00	1.00	−12.06–12.06		0.00	1.00		−6.91–6.91
Months (March to December vs. January–February)		−0.88	0.75	−6.27–4.51		−0.88	0.57		−3.97–2.21
Adjusted R-Squared		0.11				−0.03			
F-Statistics *p* value		0.01				0.87			
(**b**)									
	**Entire COVID-19 Period**			**COVID-19 Pulse Period (March to April)**			**COVID-19 Pulse Period (May to December)**		
	**Rate Ratio**	**95% CI**	***p* Value**	**Rate Ratio**	**95% CI**	***p* Value**	**Rate Ratio**	**95% CI**	***p* Value**
(Intercept)	0.00	0.00–0.00	<0.01 *	0.00	0.00–0.00	<0.01 *	0.00	0.00–0.00	<0.01 *
Time elapsed	1.00	1.00–1.01	0.14	1.01	1.00–1.01	<0.01 *	1.01	1.01–1.02	<0.01 *
COVID-19 (March to December)	1.70	1.25–2.32	<0.01 *						
Individual COVID-19 pulse period									
March–April				3.38	2.62–4.30	<0.01 *			
May–December ^1^							0.88	0.59–1.29	0.53

* Statistically significant at <0.01; ^1^ No individual bi-monthly COVID-19 pulse period was significant.

**Table 2 ijerph-19-14348-t002:** Characteristics of Deceased in Years 2019 and 2020 by Time Periods and Cause of Death.

				Deceased during 2019–2020 (*n* = 496)			Deceased during March–December 2020 (*n* = 250)		
	All-Cause in 2019	All-Cause in 2020	*p* Value ^2^	COVID-19-Specific	Non-COVID-19-Specific	*p* Value ^2^	COVID-19-Specific	Non-COVID-19-Specific	*p* Value ^2^
Characteristic	*n* (%)	*n* (%)		*n* (%)	*n* (%)		*n* (%)	*n* (%)	
**Total**	251	245		85	411		85	160	
Age at 9/11, years									
20–44	52 (20.72)	61 (24.90)	0.096	23 (27.06)	90 (21.90)	0.028 *	23 (27.06)	38 (23.75)	0.123
45–64	132 (52.59)	138 (56.33)		52 (61.18)	218 (53.04)		52 (61.18)	86 (53.75)	
≥65	67 (26.69)	46 (18.78)		10 (11.76)	103 (25.06)		10 (11.76)	36 (22.50)	
Gender									
Female	111 (44.22)	98 (40.00)	0.341	29 (34.12)	180 (43.80)	0.100	29 (34.12)	69 (43.13)	0.171
Male	140 (55.78)	147 (60.00)		56 (65.88)	231 (56.20)		56 (65.88)	91 (56.88)	
Race/ethnicity									
Non-Hispanic White	124 (49.40)	110 (44.90)	0.335	29 (34.12)	205 (49.88)	0.009 *	29 (34.12)	81 (50.63)	0.023 *
Non-Hispanic Black	49 (19.52)	62 (25.31)		30 (35.29)	81 (19.71)		30 (35.29)	32 (20.00)	
Hispanic	36 (14.34)	28 (11.43)		12 (14.12)	52 (12.65)		12 (14.12)	16 (10.00)	
All Other	42 (16.73)	45 (18.37)		14 (16.47)	73 (17.76)		14 (16.47)	31 (19.38)	
Marital Status ^1^									
All other	206 (82.07)	198 (81.48)	0.865	73 (85.88)	331 (80.93)	0.282	73 (85.88)	125 (79.11)	0.195
Never married	45 (17.93)	45 (18.52)		12 (14.12)	78 (19.07)		12 (14.12)	33 (20.89)	
ZCTA proportion of essential workers ^3^									
≤median	124 (49.40)	96 (37.96)	<0.010 *	24 (28.24)	193 (46.96)	<0.002 *	24 (28.24)	69 (43.13)	0.022 *
>median	127 (50.60)	152 (62.04)		61 (71.76)	218 (53.04)		61 (71.76)	91 (56.88)	
Income ^1^									
≥$75,000	64 (27.83)	71 (31.00)	0.455	21 (25.93)	114 (30.16)	0.448	21 (25.93)	50 (33.78)	0.219
<$75,000	166 (72.17)	158 (69.00)		60 (74.07)	264 (69.84)		60 (74.07)	98 (66.22)	
Smoking status ^1^									
Never	79 (41.15)	88 (45.60)	0.378	33 (45.83)	134 (42.81)	0.641	33 (45.83)	55 (45.45)	0.959
Ever	113 (58.85)	105 (54.40)		39 (54.17)	179 (57.19)		39 (54.17)	66 (54.55)	
History of PTSD									
No	172 (68.53)	178 (72.65)	0.313	60 (70.59)	290 (70.56)	0.996	60 (70.59)	118 (73.75)	0.597
Yes	79 (31.47)	67 (27.35)		25 (29.41)	121 (29.44)		25 (29.41)	42 (26.25)	
History of chronic health condition									
No	54 (21.51)	48 (19.59)	0.596	7 (8.24)	95 (23.11)	0.002 *	7 (8.24)	41 (25.63)	0.001 *
Yes	197 (78.49)	197 (80.41)		78 (91.76)	316 (76.89)		78 (91.76)	119 (74.38)	
Type of enrollees									
Community members	182 (72.51)	170 (69.39)	0.444	54 (63.53)	298 (72.51)	0.097	54 (63.53)	116 (72.50)	0.147
Rescue/recovery workers	68 (27.49)	75 (30.61)		31 (36.47)	113 (27.49)		31 (36.47)	44 (27.50)	

^1^ Numbers do not add up to total due to missing category. ^2^ Chi-Square test. ^3^ Neighborhoods of at least two of the following occupations are greater than median: (1) healthcare support, (2) protective service, (3) food preparation and serving related, (4) building and grounds cleaning and maintenance, (5) personal care and service, and (6) production, transportation, and material moving. * Statistically significant at <0.05.

**Table 3 ijerph-19-14348-t003:** Multinomial Analysis of the Association of Socio-economic, Behavioral and Neighborhood Factors with Cause of Death During 2019–2020 ^1^.

	Cause-Specific Death during March–December 2020 ^1^							
	Adjusted for Individual Level Factors				Adjusted for Individual and Neighborhood Factors			
	Non-COVID-19 Death (*n* = 160)		COVID-19 Death (*n* = 85)		Non-COVID-19 Death (*n* = 160)		COVID-19 Death (*n* = 85)	
	AOR	95% CI	AOR	95% CI	AOR	95% CI	AOR	95% CI
Age at Death, by year	1.00	0.98–1.02	0.98	0.96–1.00	1.00	0.98–1.02	0.98	0.96–1.01
Gender								
Female	Ref	Ref	Ref	Ref	Ref	Ref	Ref	Ref
Male	1.01	0.65–1.57	1.35	0.76–2.40	1.01	0.65–1.57	1.34	0.76–2.39
Race								
Non-Hispanic White	Ref	Ref	Ref	Ref	Ref	Ref	Ref	Ref
Non-Hispanic Black	1.08	0.61–1.92	2.43	1.23–4.77 *	0.97	0.54–1.76	1.98	0.98–4.01
Hispanic	0.73	0.36–1.48	1.56	0.67–3.64	0.72	0.36–1.46	1.48	0.63–3.48
All Other	1.19	0.65–2.15	1.72	0.78–3.78	1.13	0.62–2.06	1.54	0.69–3.43
Marital Status								
All other	Ref	Ref	Ref	Ref	Ref	Ref	Ref	Ref
Never married	1.17	0.68–1.99	0.73	0.35–1.54	1.20	0.70–2.05	0.76	0.36–1.61
ZCTA proportion of at least two of the six types of essential workers ^2^								
≤Median	―	―	―	―	Ref	Ref	Ref	Ref
>Median	―	―	―	―	1.37	0.88–2.13	1.78	0.98–3.23
Income								
≥$75,000	Ref	Ref	Ref	Ref	Ref	Ref	Ref	Ref
<$75,000	0.76	0.46–1.25	1.08	0.56–2.08	0.74	0.45–1.22	1.05	0.54–2.02
Unknown	0.65	0.27–1.59	0.72	0.20–2.60	0.67	0.28–1.63	0.76	0.21–2.76
Smoking status								
Never	Ref	Ref	Ref	Ref	Ref	Ref	Ref	Ref
Ever	0.82	0.51–1.32	0.87	0.49–1.55	0.85	0.53–1.38	0.93	0.52–1.66
Unknown	0.92	0.52–1.64	0.63	0.29–1.35	0.95	0.53–1.68	0.66	0.30–1.43
History of PTSD								
No	Ref	Ref	Ref	Ref	Ref	Ref	Ref	Ref
Yes	0.87	0.54–1.39	0.73	0.41–1.30	0.87	0.54–1.38	0.72	0.40–1.30
History of chronic health condition								
No	Ref	Ref	Ref	Ref	Ref	Ref	Ref	Ref
Yes	0.80	0.49–1.33	2.86	1.19–6.88 *	0.80	0.48–1.33	2.79	1.16–6.74 *
Type of enrollees								
Community Members	Ref	Ref	Ref	Ref	Ref	Ref	Ref	Ref
Rescue/recovery workers	1.02	0.61–1.71	1.18	0.64–2.19	0.99	0.59–1.66	1.12	0.61–2.08

Ref, referent. ^1^ Compared with all deceased during January 2019–February 202. ^2^ Types of essential workers include healthcare support, protective service, food preparation and serving related, building and grounds cleaning and maintenance, personal care and service, and production, transportation, and material moving (reference [27]). * Statistically significant at <0.05.

## Data Availability

World Trade Center Health Registry Data may be made available following review of applications to the Registry from external researchers. Mortality data may be requested from NYC DOHMH Bureau of Vital Statistics. The data are not publicly available due to privacy or ethical restrictions.

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
