# Peer review of "COVID-19-Specific Mortality among World Trade Center Health Registry Enrollees Who Resided in New York City"

_ijerph, 2022, doi:10.3390/ijerph192114348_

Round 1

Reviewer 1 Report

Thank you for the opportunity to review the manuscript entitled "COVID-19 associated Mortality among World Trade Center Health Registry Enrollees Who Resided in New York City". The authors conducted an interesting and relevant study. However, I believe in its current form the article presents significant conceptual and methodological limitations. Below, I offer some comments that I hope the authors find useful.

1. (Lines 45-48). Minor point: Depression, anxiety, and PTSD are not mental health "related" conditions, they are "mental health conditions", so the use of the word related is not appropriate (please eliminate). Major point: There is not conclusive evidence regarding psychiatric disorders association with excess morality during the pandemic (see, for example, doi:10.1016/j.lanepe.2021.100228), and indeed the authors do not provide any reference supporting their argument that "mental health conditions were also reported to increase during the pandemic and contributed to excess mortality".

2. (Lines 60-67). It is not clear why the authors chose "proportion of residents who are essential workers" as the only neighborhood risk factor. Because the authors do not provide more context, this paragraph seems isolated and unrelated to the two previous paragraphs. I would assume that the authors intended to make the point that both individual and neighborhood characteristics are risk factors, which is a good point and in fact the case for many outcomes (including COVID-19 related ones), but there are plenty of other neighborhood characteristics that can be considered a risk factor (e.g., overall population density, availability of healthcare facilities, etc). The authors need to provide a sound argument for why they are introducing neighborhood characteristics in their manuscript, and why they are including this one particular characteristic. The authors also state "Living in an area with a high proportion of 63 essential workers could result in higher COVID-19 mortality [15, 16]". This is an overstatement. Reference [15] did not include proportion of essential workers in the models examining COVID-19 deaths as an outcome (authors in this paper included this variable only for COVID-19 cases). Reference [16] only looks at correlation between specific occupations and mortality, but no regression analyses adjusting for other confounders were performed.

3. (Lines 123-145). Ascertainment of death. If I understood correctly, the authors had data to validate their matching algorithm. Since WTCHR data were periodically linked to the NYC BVS for mortality surveillance through 2018, the authors could have done the same matching process for years 2015-2018, and compare their matches to those reported in WTCHR. The WTCHR matching algorithm might not be itself 100% accurate, but the authors could have at least said that their matching algorithm for 2019-2020 was just as good as the WTCHR one. Without this cross-validation, it is hard to ascertain the reliability of the authors' matching algorithm. Minor point: there is not justification for why the 60/30-day rule was applied in June 2020 and August 3, 2020, respectively (this detail needs clarification).

4. (Lines 183-190). Excess deaths. The method used by the authors to calculate excess deaths (differences in differences) is extremely limited. By definition, excess deaths during COVID are those above the number of expected deaths in absence of the pandemic. The authors' extrapolation method implies that expected monthly deaths in 2020 were an average of observed monthly deaths in the previous 5 years, but this method is unlikely to capture any yearly trend and any other unobserved changing structure of WTCHR enrollees. In addition, their diff-in-diff coefficient estimate is most likely not significant because it is averaging changes across many months in 2020 (the DID interaction is year (2020 vs 2015-2019) times month (March-Dec vs Jan-Feb)). The following article is an excellent reference for different methods used to estimate excess deaths https://doi.org/10.1016/j.envres.2022.113754.

5. (Lines 232-239). The comparison for COVID-19 associated deaths for the entire period (2019-2020) does not seem correct. There cannot be COVID-19 associated deaths prior in 2019, so it is unclear which group is being used as the reference category. Are the authors comparing COVID-19 associated deaths in 2020 to all-cause deaths in 2019? What does the "Non-COVID-19" column refers to? In addition, the authors state "Being non-Hispanic Black and having a history of at least one chronic health condition were independently associated with COVID-19-associated deaths when evaluating the entire time period (2019-2020) and since pandemic (all p-values < 0.05)." How did the authors conclude that the significant difference for race/ethnicity was driven by non-Hispanic Black enrollees? For those deceased since the pandemic for example, about the same proportion of non-Hispanic Black enrollees are in the COVID-19/non-COVID-19 groups. Why did the authors choose to present percentages by row rather than by column? For example, intuitively, the reader would expect that the percentages would add to 100% when adding White/Black/Latinx/Other instead of by adding all Whites. 

6. (Table 3). Why did the authors not adjust for type of enrollee (rescue/recovery worker vs Manhattan community member)? 

7. (Lines 259-303). Why did the authors choose to focus their discussion of the lower mortality among WTCHR enrollees compared to the overall NYC population. This was not described as their study goal. The authors described that their aim was to examine whether there was excess all-cause and COVID-19-associated mortality among WTCHR enrollees. So, their discussion must focus on their findings related to pre/during COVID mortality rather than introducing a new aim in the discussion.

8. (Lines 305-318). This discussion about results from their WTCHR sample versus overall NYC continues on these lines. The reader would have expected to see a discussion of their pre/during COVID mortality analyses. Also, discussing this point is not aligned with the subsection title "Racial Disparity and Neighborhood Proportion of Essential Workers". 

9. (Lines 325-328). "Our study supports the finding from NYC and in other settings that living in neighborhoods with a higher proportion of essential workers is one main contributing factor to racial inequities in COVID-19-associated 327 mortality [15, 16]". References 15 and 16 cannot be used to support this finding. Reference 15 did not include proportion of essential workers when using mortality as an outcome, and reference 16 only look at simple correlations across ALL racial/ethnic groups. In addition, the authors are implicitly assuming that the non-Hispanic Black enrollees in their sample were either more likely to be employed in essential occupations or to live in neighborhood with high proportion of essential workers, but there is no supporting data included in the manuscript.

Reviewer 2 Report

This is a review of the manuscript “COVID-19 associated Mortality among World Trade Center 2 Health Registry Enrollees Who Resided in New York City” submitted for publication in International Journal of Environmental Research and Public Health. This is a very interesting and well-executed study. The methods are sound and the manuscript is well written. I have no major comments.

My first suggestion would be to examine in additional analyses the main associations while adjusting for individual chronic health conditions (including also history of major depression, which may give complementary information to PTSD) instead of at least one condition. My second suggestion would be to also examine in this sample the potential effect of the most frequently prescribed medications, if available, with mortality. These analyses would allow to examine the robustness of the findings and may enrich the discussion.

Reviewer 3 Report

Thank you for the opportunity of reviewing the manuscript about the impact of COVID-19 pandemic on the patient outcome with the WTCHR dataset. I reviewed this manuscript with great interest, but there are some problems for publication.

First, the status of COVID-19 infection in New York is not described. In many countries around the world, the number of COVID-19 infections as well as the number of deaths due to COVID-19 has generally increased over time due to an increase in the number of infected persons and the occurrence of mutant strains. In order to accurately interpret the results of this study, the relationship between the number of deaths and the number of people infected with COVID-19 should be evaluated after showing the number of people infected with COVID-19.

Next, it should be indicated why multinomial regression analysis was used in this analysis. In this study, the number of infections and deaths per unit population were evaluated as outcomes, but in studies using registry dataset, the binary variable such as death is generally evaluated as an outcome using a logistic regression model. Therefore, the authors should explain in detail the reasons for the choice of outcome and analysis method.

Finally, table 3 shows the results of the multinomial regression analysis, but the results on the left side and the results on the right side should be detailed. Perhaps it shows the difference in results due to the difference in multinomials, but as it is, the reader cannot accurately evaluate the results. Therefore, the authors should revise the table to make it easier for readers to understand.

Round 2

Reviewer 1 Report

I appreciate the authors responsiveness to my previous comments, all of which were fully and thoroughly addressed.

Reviewer 3 Report

I checked the revised manuscript, and the authors sincerely revised the problems this reviewer pointed out.

Thank you and Good luck.